# A note on variable susceptibility, the herd-immunity threshold and modeling of infectious diseases

**Marcus Carlsson** [1] *, **Jens Wittsten** [2], **Cecilia Söderberg-Nauclér** [3,4,5]

**1** Centre for Mathematical Sciences, Lund University, Lund, Sweden, **2** Department of Engineering, University of Borås, Borås, Sweden, **3** Department of Medicine, BioClinicum, Karolinska Institutet, Solna, Sweden, **4** Department of Neurology, Karolinska University Hospital, Stockholm, Sweden, **5** Institute of Biomedicine, MediCity Research Laboratory, University of Turku, Turku, Finland

* marcus.carlsson@math.lu.se

**Data Availability Statement:** All relevant data are within the paper and its Supporting information files.

## Abstract

The unfolding of the COVID-19 pandemic has been very difficult to predict using mathematical models for infectious diseases. While it has been demonstrated that variations in susceptibility have a damping effect on key quantities such as the incidence peak, the herd-immunity threshold and the final size of the pandemic, this complex phenomenon is almost impossible to measure or quantify, and it remains unclear how to incorporate it for modeling and prediction. In this work we show that, from a modeling perspective, variability in susceptibility on an individual level is equivalent with a fraction $\theta$ of the population having an "artificial" sterilizing immunity. We also derive novel formulas for the herd-immunity threshold and the final size of the pandemic, and show that these values are substantially lower than predicted by the classical formulas, in the presence of variable susceptibility. In the particular case of SARS-CoV-2, there is by now undoubtedly variable susceptibility due to waning immunity from both vaccines and previous infections, and our findings may be used to greatly simplify models. If such variations were also present prior to the first wave, as indicated by a number of studies, these findings can help explain why the magnitude of the initial waves of SARS-CoV-2 was relatively low, compared to what one may have expected based on standard models.

## 1 Introduction

Since the fundamental works of Kermack and McKendrick [1–3] compartmental mathematical models (such as SIR, SEIR, etc.) are used to model the spread of infectious diseases. Among other things, these papers introduced the by now famous $R_0$-value and showed that, in contrast with human intuition, an infectious disease will never infect the whole population, no matter how infectious. Instead, the incidence will start to decay when the fraction of recovered reaches the so called the "Herd-Immunity Threshold", for which they deduced the famous formula

$$1 - 1/R_0. \tag{1}$$

**Funding:** The research of J. W. was supported by the Swedish Research Council (2019-04878). The research of C. S-N. was supported by the Swedish Medical Research Council (2019-01736) and Flagship InFLAMES, Finland.

**Competing interests:** The authors have declared that no competing interests exist.

However, prior to the SARS-CoV-2 pandemic, there was no reliable data from a novel virus (affecting humans) on which this prediction could be tested. Unfortunately, this remains largely the case, since e.g. lockdowns and voluntary isolation (which the models can not predict) had a major effect on the spread. Despite this, data from places like Sweden, that did relatively little to stop community transmission, indicate that the mathematical models have a tendency to overestimate the magnitude of the wave during a major outbreak [4].

Several factors are known to have a damping effect on model curves. One such example is variable susceptibility, see e.g. Ch. 1 and 3 in [5], and the articles [6–9]. By variable susceptibility we here refer to (time-invariant) differences between individuals in the probability of becoming infected, given a certain exposure to the virus, as opposed to individual variations over time. Similar results have also been established numerically for other heterogeneities, such as age and activity [10]. Curiously, variable infectivity (super-spreaders) do not have any damping effect on the spread during a major outbreak [11]. In any case, such conclusions are derived using heuristic arguments or by simply testing relevant models, and the mechanisms behind these phenomena remain poorly understood. In particular, since variability in susceptibility is virtually impossible to quantify, it is unclear how to efficiently incorporate it into the models, wherefore predictions of future COVID-19 waves, or the next pandemic, continues to be a major challenge.

Concretely, suppose a novel infectious disease, whose transmission dynamics involves high variability in infectivity and/or susceptibility, is introduced in a well connected network like a large city, and suppose a major outbreak is about to take place. One may then estimate $R_0$, i.e. a rough estimate of the average number of new infections that one infective gives rise to, from the data series of early cases, using e.g. EpiEstim [12] or [13]. By a contact tracing study one may also estimate the generation time $T_{generation}$, which is the other parameter needed to run a SIR-model. In such a scenario, one can ask the question if the output of a simple SIR-simulation is a good first order approximation of what is about to come, in the absence of Non-Pharmaceutical Interventions? Is the formula (1) a good indicator of when we may expect the outbreak to start to recede?

Based on data from Sweden during the COVID-19 pandemic, the answer seems to be no, see [4] where it is shown that the incidence dropped, unexpectedly, at levels of sero-prevalence much lower than predicted by (1). Of the prior theoretical studies on this topic, the article that comes closest to answering the above questions is Britton et. al. [10], where the authors prove that variations in activity patterns can significantly lower the herd-immunity threshold, in comparison with the *classical* estimate based on (1). An older publication with a similar message is [14]. However, these conclusions are empirical observations based on models which have been built to incorporate population heterogeneity. This damping effect has not been established mathematically and it remains unclear how, and to what extent, different heterogeneities are manifested. In particular it remains unclear how to more accurately predict the herd-immunity threshold. We remark that, in the case of SARS-CoV-2, a number of factors such as genetic, cross-reactive immunity and innate immunity, have been shown to provide variation in susceptibility [15–18].

## 1.1 Novel contributions

In this work we prove mathematically that variations in susceptibility have a damping effect on the model curves, whereas variations in infectivity do not (as long as it is uncorrelated with the former, see [7]). More importantly, we also find that the (usually unknown) distribution describing *how* susceptibility varies is not needed for accurate modeling. More precisely we show that a susceptibility heterogeneous model will behave almost identically to a standard

(homogeneous) SIR-model where a portion of the population have sterilizing immunity, and that the precise shape of the susceptibility distribution only influences the level of sterilizing immunity. It is important to underline that this immunity only exists within the mathematical model simplification, and should not be confused with real sterilizing immunity of some individuals. In other words, even if everyone is susceptible to the virus (to some degree), on a population level it will seem as if a portion of the population have sterilizing immunity. We will refer to such an immunity, needed for accurate mathematical modeling, as "Artificial Sterilizing Immunity" (ASI), and the fraction of the population having it as $\theta$. As long as $\theta$ can be estimated from available data, we show that the actual Herd-Immunity Threshold is indeed lower than (1) predicts. The correct formula, in the presence of variable susceptibility, is given by

$$(1 - \theta)(1 - 1/R_0), \tag{2}$$

and the final size of the pandemic is also shrunk by the same factor $(1 - \theta)$. We shall also demonstrate numerically that other population heterogeneities, such as those considered by Britton et. al. [10], have an analogous effect, and hence the findings in this paper can be used to significantly reduce the amount of unknowns in a more realistic heterogeneous model for disease spread.

## 2 The mathematics of infectious disease spread dynamics

In order to explain the mathematical findings, we first give an overview of how the basic SIR-model works. SIR stands for Susceptibles, Infectives and Recovered, and is the simplest form of a "compartmental model" used in mathematical epidemiology (see e.g. [19] for an introduction to this field). In the model, $S$, $I$ and $R$ are functions of time $t$, and to illustrate how these are related we shall also introduce the (redundant) function $v$ describing the incidence, i.e. the amount of newly infected each day (not to be confused with $I$, which describes the prevalence). The formula for $v(t)$ is at the heart of the algorithm, and in the beginning we simply have $v(t) = \alpha I(t)$, where $\alpha$ is a constant that determines how many new cases an average infective gives rise to during a day. If $a$ is the average number of daily potentially infectious contacts by an average person, and $p$ is the probability that such a contact actually leads to transmission, then $\alpha = ap$.

As the amount of susceptibles gradually decreases, we have to modify this by multiplying with the fraction of the population that is still susceptible. If the total population is $N$ this fraction is $S(t)/N$ and the formula becomes

$$v = \frac{\alpha}{N} SI = \frac{ap}{N} SI. \tag{3}$$

To set up the remaining equations we also need the generation time $T_{generation}$, i.e. the average time it takes from infection to recovery. The remaining equations are then

$$\begin{cases} S' &= -v \\ I' &= v - \sigma I \\ R' &= \sigma I \end{cases} \tag{4}$$

where $\sigma = 1/T_{generation}$ and $'$ indicates differentiation. The equations are intuitively easy to understand, the incidence continuously gets withdrawn from $S$ and added to $I$, and at the same time there is a current of recovering individuals that leave $I$ at a rate $\sigma I$ and appear in $R$ instead.

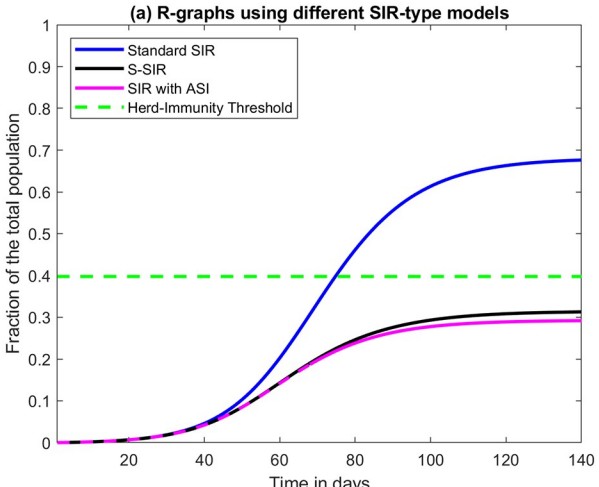
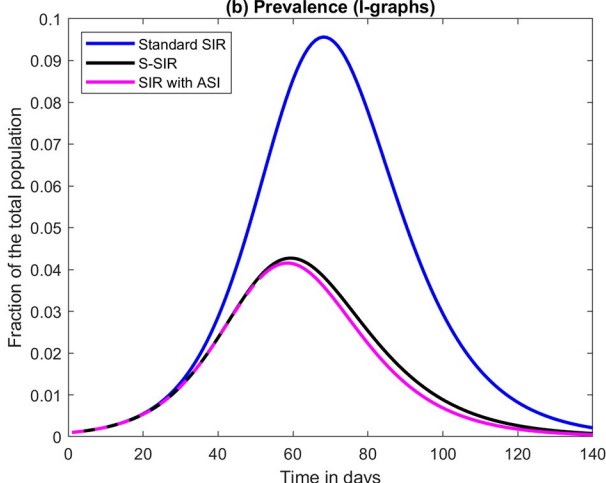

**Fig 1. Graphs of recovered *R* and prevalence *I*.** (a) Graphs of recovered (as a fraction of the total population) for various SIR-models and a fixed value of $R_0 = 1.66$. First we display standard SIR, then S-SIR and finally SIR with Artificial Sterilizing Immunity (ASI) with parameters from (8). Note that they start out almost identically but that the latter two bend downwards much earlier than the first, which over-shoots the classical Herd-Immunity Threshold (HIT), whereas the second two stay closely together and level out below the classical HIT. (b) Corresponding curves for prevalence *I* (the *S*-graphs are shown independently in Fig 2).

The SIR-model, and our extensions thereof, are deterministic in the sense that if we run it twice, the output is the same. Such models are believed to work well during major outbreaks, where the law of large numbers applies [5, 11]. All our findings pertain to this situation; for modeling of e.g. the initial phase or household transmission, other types of models are used.

The most natural initial condition for a new disease is to set $I(0) = n$ where $n << N$ represents a small number of import cases arriving at time $t = 0$, and then set $S(0) = N − n$ and $R(0) = 0$ (so everybody else is initially susceptible and no-one has yet recovered). The value of $n$ is completely irrelevant for the shape of the curves that follow, a low value of $n$ only gives the equation system a slower start so it takes a while longer for the outbreak to reach a certain value. Once this happens, the curves look exactly the same independent of the value $n$. See the blue graphs in Fig 1 for some typical examples of *R*-curves and *I*-curves. In this model, *R* is always increasing and levels out on a number which is called "the final size of the pandemic" (see Fig 1a). *S* approximatively looks like $N − R$, since the prevalence *I* at any given time is small in comparison with the total population. The incidence $v$ typically looks just like *I*, albeit with a lower magnitude.

## 2.1 Contemporary models for COVID-19

Contemporary models used by professional modeling teams usually contain many more compartments than SIR, for instance relating to age stratification, variable activity levels, geographical regions, compartments for people who need ICU and compartments for people that die. For example, the model published by members of the Imperial College COVID-19 response team [20] has at its root a basic SIR (see p. 9 as well as S2 Fig in the supplementary material of [20]), and the same goes for the model [21] used by a renowned Swedish modeling team, which managed to fit the ICU occupancy and deaths with high accuracy during the first wave in Sweden. The latter model also takes into account various regions and interaction patterns between these, but the in-region dynamics is basically a simple SEIR. It is also common to add a compartment *E* for "Exposed", incorporating the incubation time, (as indeed is done in the

above two examples). However, as we shall show in Section 4, this has a limited effect on the overall behavior. By this we mean that, for every set of parameter values ($R_0$, incubation time etc.) for SEIR, it is possible to get an almost identical curve with SIR (and vice versa), if we are allowed to alter the parameter values slightly. Since the exact value of these parameters is never known, this means that for practical purposes one may just as well rely on SIR as on SEIR, at least for understanding overall trends. For example, in Fig 3 we show an example of SEIR and SIR with $R_0$-values that differ by 1%, and the graphs are almost identical. For example the final size of the pandemic differs by less that 1.5%. Moreover, compartments relating to severely ill and deaths also have a marginal effect on the overall behavior, simply because only a small fraction of the infected will end up in these compartments. Based on this, we argue that, for the purposes of understanding the general overall behavior, as we are interested in here, it suffices to study the simpler SIR-model. For other attempts to predict/model SARS-CoV-2 using SIR/SEIR-type models see e.g. [22, 23].

In contrast, other types of heterogeneities such as variable activity levels and different interaction patterns between age groups, does have a notable damping effect on the model curves. For example, the age-activity stratified SEIR by Britton et. al. [10] has an incidence peak of about 35% lower than standard SIR, given analogous input parameters. This is consistent with the findings in [10], where a drop in the Herd-Immunity Threshold of around 30% is observed for the age-activity model, comparing with the prediction (1) based on SIR. This will be further discussed in Section 4.2. Also variable susceptibility has a major effect, but this has already been discussed in the introduction and is further analyzed in Section 3.

## 2.2 Model versus reality mismatch?

Whether or not the more advanced models accurately describe the spread of COVID-19 is hard to determine, since one may always argue that Non-Pharmaceutical Interventions (NPI's) as well as voluntary behavioral changes have had a major impact. Without claiming to have a definite answer, the case of Sweden is interesting due to its relaxed strategy, which moreover was kept almost constant during 2020–2021. In particular, schools were kept open, people who could not work from home were encouraged to go to work, family members of infected households were obliged to work or go to school, and widespread face-mask use was never implemented, making the country ideal for comparing models with actual data. Due to insufficient testing, the time series of cases is of limited value, but measurements of sero-prevalence from blood samples give valuable information, since it has been established that most people who get COVID-19 also go on to develop anti-bodies [24], and that these antibodies remain for at least 9 months [25, 26]. Results published by the Swedish Public Health Agency [27] indicate that roughly 11% had had COVID-19 in the Stockholm region after the first wave 2020, which rose to around 22% in February 2021, following the second wave. Also among hospital staff in Sweden (not using face-mask), the prevalence was around 20% [26] after the first wave, in line with observations from infected households elsewhere [28].

However, the model by Sjödin et. al., referred to earlier, predicts a cumulative number of infected people of around 30% after the first wave, despite assuming a 56% decrease in contacts among people of age 0–59 and a 98% reduction among those aged 60–79 (this is for scenario d which accurately fitted ICU-occupancy and death, see Fig 2b, bearing in mind that the Stockholm region has 2.4 million inhabitants). Along the same line, Britton et. al. [10] estimated that the disease could level out at around 43% total infected, in a matter of months. While the authors stress that this is not an actual prediction, it is based on realistic parameters for COVID-19. The famous Report 9 by the Imperial College [29] predicted a total number of

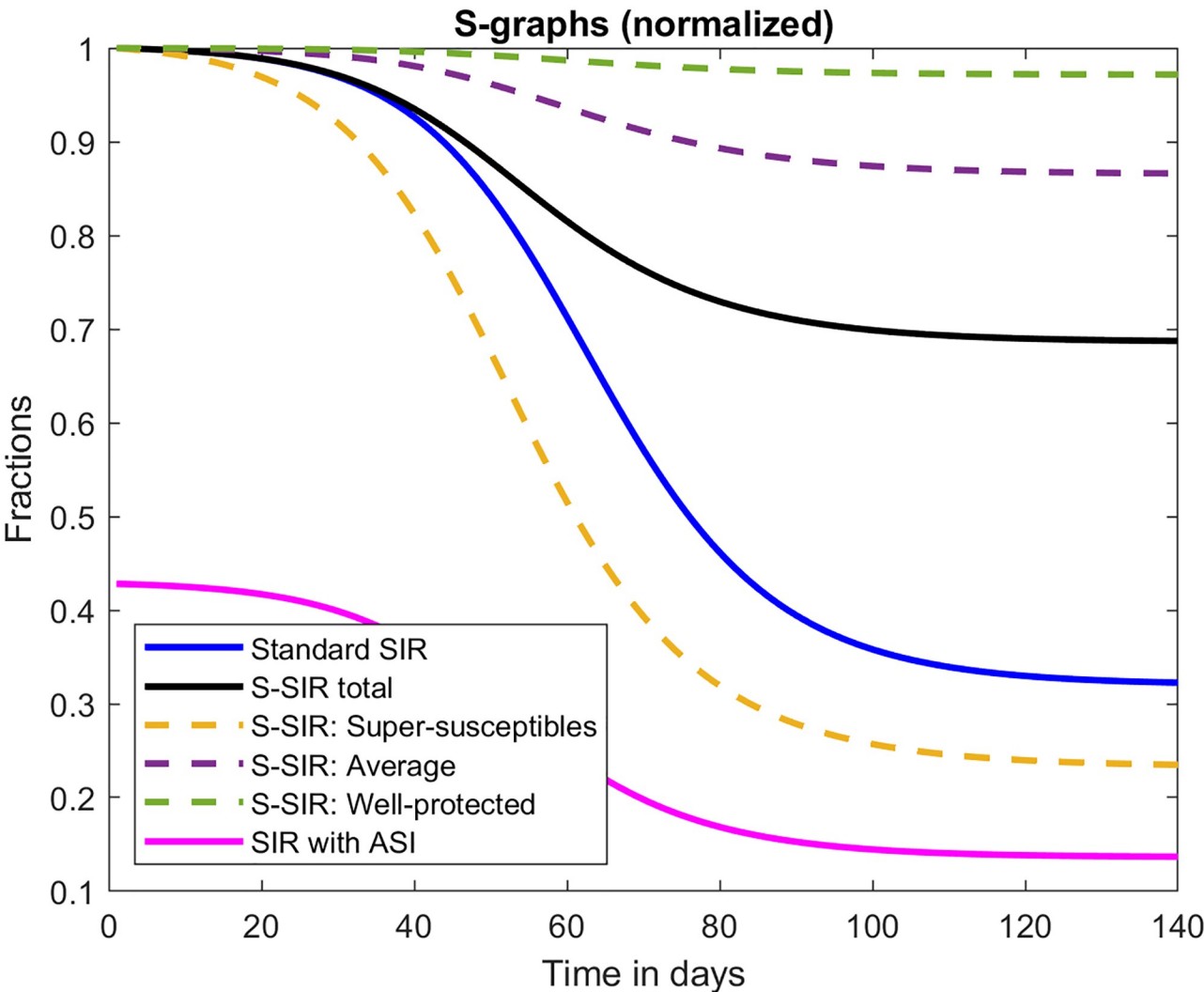

**Fig 2. Graphs of susceptibles *S*.** *S*–curves corresponding to the 3 graphs in Fig 1. As in Fig 1, the blue black and pink have been normalized by division by *N*. The black curve thus shows the proportion of the total population susceptible to the virus. Note that when the pandemic is over, around 68% are still susceptible, in stark contrast to classical SIR which levels out at around 34%. The pink curve starts out assuming 57% have artificial sterilizing immunity, and hence its initial value is 43% (this number was chosen using the formula (8)). Note that the pink curve looks exactly like the black except for a vertical translation, illustrating the key findings of this article. The S-SIR model has three subgroups $S_1$, $S_2$, $S_3$ corresponding to $p_1 = 1$ (labeled "super-susceptibles"), $p_2 = 0.1$ (labeled "normal") and $p_3 = 0.02$ (labeled "well-protected"). Here we have normalized with the amount of people in each respective subgroup, wherefore all curves start at 1. Note how the spread in the latter two sub-groups level out as soon as it levels out in the super-susceptible group.

81% infected in a "do-nothing" scenario, based on a more advanced so called "agent based model" that also treats household-contacts separately. According to Table 3 in the report, the number of deaths and peak ICU capacity can be reduced by 50% and 81%, respectively, in the most effective NPI-scenario, which certainly goes beyond what was implemented in Sweden. However, as of February 2021, when the original Wuhan-strain was declining [30], these reduced predictions overestimate the actual figure by a factor of roughly 4 (deaths) and 10 (ICU) (when directly translated to Stockholm County).

The point here is not to criticize any particular model, and clearly the case of Sweden alone can not prove that models are right or wrong, as mentioned initially. However, based

on the massive discrepancy between the actual Swedish data and the various model outcomes described above, it is a legitimate question whether "contemporary models" have a tendency to significantly overestimate the society spread and final size of the pandemic. We find it likely that the answer is yes, and further support for this hypothesis is given in [4]. In this paper we demonstrate that variable susceptibility is one factor that contributes to this phenomenon.

## 2.3 Pre-immunity, super-spreaders and other inhomogeneities

How can we alter the equation system (3) and (4) in order to dampen the curves? The simplest option is to assume that a certain fraction $\theta$ of the population have some form of sterilizing immunity so that they can not get infected by the virus. Mathematically, this is easily achieved by updating the initial conditions to

$$\begin{cases} S(0) & = \omega N \\ I(0) & = n \\ R(0) & = 0 \end{cases} \tag{5}$$

where $\omega = 1 - \theta$ is the fraction of initially susceptible. However, this is not very realistic since immunity is usually not binary, i.e. either 0% or 100% (so called sterilizing immunity). The hypothesis that some people are more susceptible than others is then far more plausible than a binary immunity. In the particular case of SARS-CoV-2, the hypothesis that certain individuals had some form of pre-immunity was suggested in various publications as an explanation for the, at least according to some, unexpectedly mild initial infection waves, see for instance [31]. This paper also lists a number of studies showing that some people had some a priori T-cell immunity. Since then, different articles have demonstrated various mechanisms that make certain individuals more or less susceptible to SARS-CoV-2, e.g. [15–18]. It is also well established that infectivity levels vary dramatically, as mentioned earlier (see e.g. [32]). In addition, this seems uncorrelated to how sick they become; many individuals with very high viral loads are even asymptomatic. In light of this, the most probable assumption is that also the way the virus enters the human is subject to large individual variations.

To make a more realistic model for the spread of COVID-19, or any infectious disease for that matter, it is reasonable to divide the compartments $S$ and $I$ into a number of subcompartments $S_1, \ldots, S_J$ and $I_1, \ldots, I_K$ where people in each compartment have a different level of susceptibility/infectivity. To see how to set up a corresponding equation system for disease spread, recall that $a$ was the amount of daily contacts by one individual. We now let $p_{jk}$ be the probability that such a contact leads to transmission when an individual in $S_j$ meets one in $I_k$. The incidence $v_j$ coming from the group $S_j$ then becomes

$$v_j = \frac{S_j}{N} \left( a p_{j1} I_1 + \ldots + a p_{jK} I_K \right) \tag{6}$$

(c.f. (3)). Since we assume no correlation between infectivity and susceptibility, the total amount of new infectives $v_1 + \ldots + v_J$ is then distributed among the groups $I_1, \ldots, I_K$ according to their relative size. The remaining equations in (4) are easily modified to this new vector setting, we refer to Sec. 1 in S1 File for the details. In the coming section we analyze the behavior of this system of equations, and in Section 4 we also discuss other extensions such as SEIR and variable activity levels.

## 3 Main results

The main point of this research is that extensions to both SIR and SEIR of the type mentioned above yield overall curves that are only marginally different from basic SIR, given that a level of Artificial Sterilizing Immunity (ASI) is included. First of all, after setting up the details in Section 1 of S1 File, we prove in Proposition 1.1 that the division of $I$ into various sub-compartments have no effect whatsoever, further supporting the conclusions in [8, 9, 11]. In other terms, the existence of "super-spreaders" do not in any notable way affect the dynamics of disease spread. Removing this layer of complexity, the Eq (6) simplify to

$$v_j = \frac{ap_j}{N} S_j I \tag{7}$$

where $p_j$ is the probability of transmission when a susceptible in group $S_j$ encounters an "average" infectious individual. We refer to Eq (14)-(16) in S1 File for the full system of equations, which we label S-SIR for "Susceptibility-Stratified SIR". It is a very curious fact that the division of $S$ into subcompartments can not, in contrast to $I$, mathematically be further reduced to a simpler equation system. However, and this is the key result of this paper, we can prove mathematically that the overall behavior of S-SIR (in terms of prevalence $I$ and recovered $R$) differs only marginally from the basic SIR (3) and (4) upon including ASI to the initial conditions, as we did in (5). This is the essence of Theorem 2.1, which is found in Section 2 of S1 File. Given probabilities $p_1, \ldots, p_J$, the theorem also provides formulas for suitable values of the transmission coefficient $\alpha$ (used to compute the incidence $v$ in (3)) and artificial sterilizing immunity $\theta$ (used in the initial conditions (5)), as follows:

$$\alpha = a \frac{\sum_{j=1}^{J} w_j p_j^2}{\sum_{j=1}^{J} w_j p_j}, \quad \omega = \frac{\left(\sum_{j=1}^{J} w_j p_j\right)^2}{\sum_{j=1}^{J} w_j p_j^2}, \tag{8}$$

where $\omega = 1 - \theta$ and $w_j$ is the fraction of the population initially belonging to $S_j$; $w_j = S_j(0)/N$. A simple illustration of these results is found in Section 1.3 in S1 File. It is important to be careful with the interpretation of $\theta = 1 - \omega$ as a fraction of people who actually have sterilizing immunity, since there is, in reality, not a division of $\theta N$ immune and $\omega N$ susceptible, which is why we have chosen the acronym ASI; *artificial* sterilizing immunity. These results are illustrated in Figs 1 and 2. Note in particular that, rather surprisingly, as soon as the most vulnerable susceptibility group (labeled super-susceptibles in Fig 2) runs out of new individuals to infect, transmission in all other groups cease as well. This behavior is typical, see S1 Fig in S1 File for a similar example with different values.

We have observed the same phenomenon also when modeling with SEIR and also when including e.g. different age groups and variable activity levels, following [10]; models with many such layers produce output which seem practically indistinguishable from the output of SIR with ASI, i.e. (3)–(5). We leave as a numerical observation which we discuss further in Section 4. In particular, given an estimated level of ASI $\theta$ in a society, it is mathematically impossible to draw any conclusions about how much of $\theta$ is caused by inhomogeneities in age and behavior, and how much comes from variations in susceptibility.

Incidentally, at the end of each paper [1–3], Kermack and McKendrick stress that a weakness in their model is that they assume uniform susceptibility, which they consider unrealistic in many cases. However, it seems that they never got around to address this issue, and we have not found a rigorous mathematical analysis of how to deal with this situation elsewhere in the literature either. In particular, the formula $1 - 1/R_0$ for the Herd-Immunity Threshold (HIT), which stems from their seminal papers, may very well be inaccurate, as suggested also in [10]. In the coming section we derive a refined version of this formula taking ASI into account.

### 3.1 Formulas for $R_0$ and the herd-immunity threshold

It is easy to see that the generation time $T_{generation}$ (introduced below (3)) coincides with the average time an infected individual remains infective. Since $\alpha$ is the infection rate, we conclude that $R_0 = \alpha T_{generation}$ for the standard SIR (3) and (4), assuming a fully susceptible population. However, in the presence of ASI $\theta$, the actual infection rate is only $(1 - \theta)\alpha$ and hence the correct formula for the $R_0$-value becomes

$$R_0 = (1 - \theta)\alpha T_{generation} = \omega \alpha T_{generation}. \tag{9}$$

The above value for $R_0$ is the value that would be estimated by e.g. EpiEstim [12] or [13] from a real time series generated by the model (3) and (4) with initial data (5). Mathematically, $R_0$ is defined as the number of new infections that one infected individual gives rise to, before disease induced immunity starts to build up. (To compute this, first solve $I'(t) = -\sigma I(t)$, given $I(0) = 1$, recalling that $\sigma = 1/T_{generation}$, and then integrate the resulting incidence $\nu$, as given by (3), while keeping $S(t)$ fixed at $S(0) = \omega N$.) Similarly, one sees that the effective $R$-value, denoted $R_e(t)$, in the above model is

$$R_e(t) = \frac{S(t)}{N}\alpha T_{generation} = \frac{S(t)}{S(0)}R_0.$$

The term "herd-immunity" carry a variety of meanings [33]. In mathematical epidemiology, given a certain model and a novel virus, the Herd-Immunity Threshold is defined as the total number of infective and recovered needed to achieve $R_e(t_0) = 1$. Since

$$I'(t) = \frac{\alpha}{N}(S(t) - \sigma)I(t) = (R_e(t) - 1)\sigma I(t),$$

(recall (4)), we see that this coincides with the point at which the wave of infectious naturally starts to recede. Beyond this point, any import cases will not spark new outbreaks. We denote this value by $H_{IT}$.

In the SIR-model, it is assumed that individuals mix homogeneously and that recovered individuals have protective antibodies (i.e. sterilizing immunity). While it is known that antibodies wane over time, at least for SARS-CoV-2, this waning happens much more slowly than the duration of an outbreak [25], and hence the latter assumption is reasonable for the discussion of the herd-immunity threshold in a shorter time frame. However, we wish to stress that the waning means that herd-immunity is never a stable condition, but will fade with time, and hence the fact that herd immunity is reached during a particular wave does not prevent future waves, which may occur either due to waning antibodies or the emergence of new variants.

Assume now that a SIR-model with a certain level of ASI accurately describes a given outbreak. The Herd-Immunity Threshold $H_{IT}$ then equals $S(0)/N - S(t_0)/N$ where $t_0$ is the time point when the herd-immunity threshold is reached, which can be found by solving $R_e(t_0) = 1$. In other words $H_{IT}$ is the difference between the fraction $S(t_0)/N$ of susceptibles at the time $t_0$ when herd-immunity is reached, and the fraction of susceptibles initially. In the SIR-model with ASI, solving $R_e(t_0) = 1$ yields the equation $S(t_0)/N = 1/\alpha T_{generation}$, and so we deduce

$$H_{IT} = \omega - \frac{1}{\alpha T_{generation}} = \omega(1 - 1/R_0), \tag{10}$$

where we used the earlier formula (9) as the definition of $R_0$. This is the formula for the herd-immunity threshold presented in Eq (2) in the introduction. It implies that the classical formula (1), given an estimate of $R_0$ from e.g. EpiEstim, is over-estimating the herd-immunity

threshold. More importantly, it allows to predict $H_{IT}$, given that the ASI parameter $\theta = 1 - \omega$ can be estimated from available data.

That the classical formula may be misleading has been pointed out before [14], and a more recent contribution indicating that the $H_{IT}$ could be significantly lower than the value (1) is [10]. These works illustrate this by simply testing models that involve heterogeneities (primarily social mixing patterns, not variable susceptibility), and therefore it offers little guidance for actual estimation of $H_{IT}$. Formula (2) is, to our knowledge, the first time this effect has been given a mathematical formula.

To sum up, we have deduced a new formula for the herd-immunity threshold in the model SIR with ASI. Since the results in Section 3 imply that this is a good approximation to Susceptibility-stratified SIR, it follows that the above formula applies to this model as well, with $\omega$ given by (8). In Section 4 we demonstrate numerically that the same conclusion seems to be true also for other heterogeneities, and hence the formula may be a better alternative for estimating the herd-immunity threshold more generally (assuming that the value of $\theta$ can be inferred from available data).

It is crucial to note that (10) applies under the assumption that the immunity is achieved by natural spread. The herd-immunity threshold for vaccinating is still given by the classical formula (1) (assuming the vaccine gives sterilizing immunity), which is shown in Section 1.2 of S1 File. This indicates that it is harder to achieve herd-immunity by vaccination, but more work is needed to establish these results in practice.

### 3.2 Damping and the final size of the pandemic

As mentioned earlier, several works have established that variable susceptibility have a damping effect on the prevalence. By the above results, this can now can be quantified. Suppose $(\tilde{S}, \tilde{I}, \tilde{R})$ is a solution to SIR in a homogenous and fully susceptible population (so $\tilde{S}(0) = N$), and let $\tilde{\alpha}$ be the corresponding transmission rate. Given a fixed value of ASI $\theta$, it is then easy to see that $(S, I, R) = (\omega\tilde{S}, \omega\tilde{I}, \omega\tilde{R})$ is a solution to (3)–(5), where $\omega = 1 - \theta$ and $\alpha = \tilde{\alpha}/\omega$. Hence the effect of ASI is really nothing but a rescaling of standard SIR curves. Note that rescaling does not change the value of $R_0$, which due to formula (9) is given by $\omega\alpha T_{generation} = \tilde{\alpha} T_{generation}$ in both cases.

It is well known that the final size of the pandemic $\tilde{\pi} = \tilde{R}(\infty)/N$ in the usual SIR (as well as SEIR) is given by solving $1 - \tilde{\pi} = e^{-R_0\tilde{\pi}}$ (see [9] and Chapter 3 of [5]). Combining this with the above we see that the final size of the pandemic $\pi$ in SIR with ASI is given by solving

$$1 - \pi/\omega = e^{-R_0\pi/\omega}.$$

Hence, in combination with our main result about reduction of Susceptibility-Stratified SIR to SIR with ASI, we deduce that the above solution $\pi$ is a good approximation to the final size of the pandemic for S-SIR with $\omega$ given by (8).

## 4 Extension to more general models

For a disease like COVID-19, with a short incubation period followed by an even shorter infectious period, there is only a marginal difference between modeling using SIR and using SEIR, and hence we believe that the key conclusions of this paper extend to this model as well. Similarly, we have found numerically that more advanced SEIR-models taking variable age and activity levels into account, behave just like SIR if we incorporate ASI. We leave the formal verification of these observations as an open conjecture, and content ourselves with showing some examples.

## 4.1 SEIR

SEIR has two key parameters apart from $R_0$, namely $T_{infectious}$ and $T_{incubation}$, where the former is the average time that a person is infectious and the latter is the time from when a person becomes infected until he or she becomes infectious. Estimates for these vary, we here follow Britton et. al. [10] and set $T_{incubation} = 4$ and $T_{infectious} = 3$. It then follows that the *generation time* equals

$$T_{generation} = T_{infectious} + T_{infective} = 7,$$

where the generation time is the average time it takes from a person getting infected until that person infects others (see Eq (5) in the supplementary material to [30] for a formal derivation). Note that this is consistent with the choice of $T_{generation}$ in previous sections.

The reason why SEIR and SIR give almost identical output for COVID-19 is that both are primarily determined by the values of $T_{generation}$ and $R_0$. To wit, during a major outbreak, it does not matter if a person is sick for 7 days and infect $R_0$ people during those 7 days, or if he undergoes incubation for 4 days and then infect $R_0$ people during the remaining 3 days. As an example, consider Fig 3(a); we see a very similar behavior by choosing parameters for SIR and SEIR in accordance with the above formulas (with $R_0$ fixed). Moreover, by allowing free parameters, SIR can be made to behave almost identically as SEIR (even without involving ASI). To support this claim, not the almost perfect overlap between the blue and black curves in Fig 3, obtained by keeping $T_{generation}$ fixed and modifying $R_0$ by one percent. Since the exact value for the input parameters are unknown in reality, we argue that it is irrelevant whether one uses SIR or SEIR, at least for modeling of SARS-CoV-2 and viruses with similar characteristics. Therefore, the observations of this paper should extend to SEIR as well, even if we have not been able to establish this mathematically.

## 4.2 Heterogeneous models

Variable susceptibility is not the only type of population heterogeneity which could manifest itself as ASI on a macro level. In [10] the authors develop a heterogeneous SEIR model taking

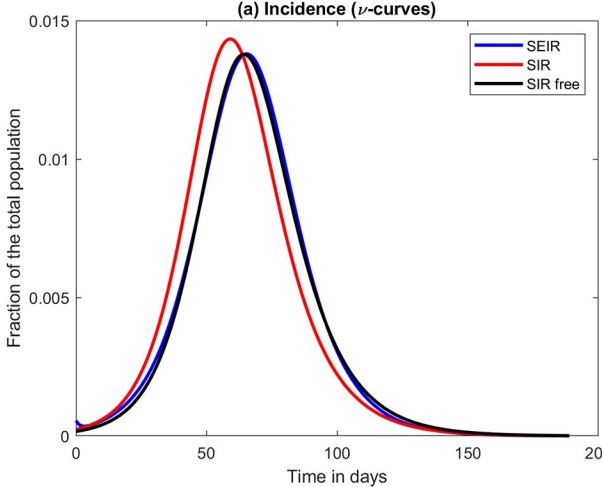
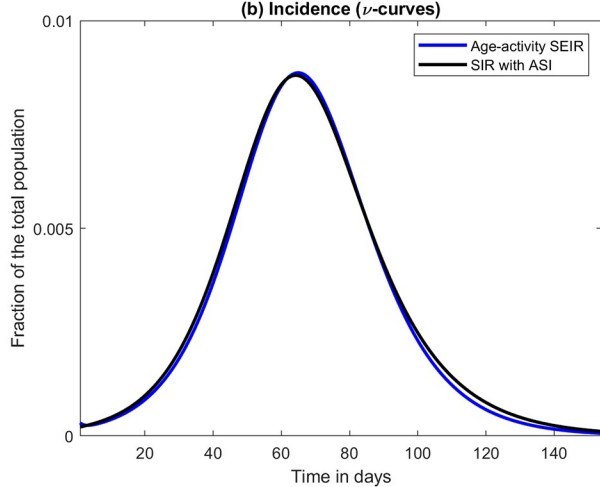

**Fig 3. Approximations using SIR with ASI.** (a) SEIR with $R_0 = 1.66$ and $T_{infectious} + T_{infective} = 7$ (blue), SIR with the same $R_0$ and $T_{generation} = 7$ (red) and finally SIR with a 1% lower $R_0$, same $T_{generation}$ (black). (b) Age-activity stratified SEIR with $R_0 = 1.66$ and $T_{infectious} + T_{infective} = 7$ (blue); SIR using the same $T_{generation}$ but an ASI of 25% and slightly different $R_0$ (black).

variable interaction pattern between different age-groups into account, as well as the fact that people in each age-group have varying amount of contacts. We implemented their model and then sought parameters for SIR with ASI that would yield a similar output. The result is seen in Fig 3(b). Again, the difference is so fine that it would be impossible to spot in practice. Henceforth, what may appear as a certain level of population (pre-)immunity in mathematical models may in fact be a mix of various population heterogeneities, in which variable susceptibility is only one ingredient.

## 5 Discussion

There could be many reasons for why certain people are more susceptible than others to infection by a novel virus, ranging from innate and adaptive immunity to cross-reactive immunity from other known viruses as well as genetic differences. For a novel disease, sterilizing pre-immunity, i.e. individuals which are completely immune without ever having had the virus, most likely does not exist. The key point of this study is that sterilizing individual immunity is not needed in order to observe what looks like sterilizing immunity on a population level, which we have coined ASI; artificial sterilizing immunity. We show mathematically that, in order to have ASI, we only need moderate variation in susceptibility. Moreover, we demonstrate numerically that other types of population heterogeneities, such as variable social mixing patterns, also manifest themselves as ASI. The findings in this paper do not limit themselves to SARS-CoV-2, but basically shows that classical formulas for the herd-immunity threshold and the models for spread of infectious diseases with roots in the famous paper by Kermack and McKendrick [1] are inapt to model any infectious disease subject to large variability in susceptibility, and need to be modified as described in Section 3.1.

The estimation of the herd-immunity threshold $H_{IT}$ is crucial for efficient disease control management and planning. For example, if a society decides to make a lock-down before $H_{IT}$ is reached, it is almost certain that the disease will re-emerge unless NPI's are maintained indefinitely. The classical formula (1) is still very much in use, despite the fact that it is known to rely on a number of oversimplifying assumptions which may lead to an erroneous indication. We have established a new formula which we prove applies when variable susceptibility is present. Since we show that our simplified model, SIR with ASI, also seems to be a good substitute for models that involve variable social mixing patterns, it is possible that (2) applies more generally than what we are able to prove mathematically.

## Supporting information

**S1 Data.**
(ZIP)

**S1 File. Supplementary material.**
(PDF)

## Acknowledgments

We thank Erik Wahlén for fruitful discussions.

## Author Contributions

**Conceptualization:** Cecilia Söderberg-Nauclér.

**Formal analysis:** Marcus Carlsson, Jens Wittsten.

**Investigation:** Marcus Carlsson, Jens Wittsten.

**Methodology:** Marcus Carlsson.

**Writing – original draft:** Marcus Carlsson, Jens Wittsten.

**Writing – review & editing:** Jens Wittsten, Cecilia Söderberg-Nauclér.

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
