## [Decision Letter · Decision Letter 0]

8 Apr 2022

PONE-D-22-06269The role of variable susceptibility and infectivity in the spread of SARS-CoV-2PLOS ONE

Dear Dr. Carlsson

Thank you for submitting your manuscript to PLOS ONE. After careful consideration, we feel that it has merit but does not fully meet PLOS ONE’s publication criteria as it currently stands. Therefore, we invite you to submit a revised version of the manuscript that addresses the points raised during the review process.

As pointed out by reviewer 2, it is far from clear how the authors monitored the seroconversions , how they  occured and what was the proportion pertaining to innate immunity. This is a critical point which needs to be clari ,authors should explain what they mean by herd immunity, a highly debated subject with little solid data.

.in addition the mathematical model to account for the epidemy and its evolution  appears to be highly simplified

We look forward to receiving your revised manuscript.

Kind regards,

Jean-Luc EPH Darlix, MG, Ph.D.

Academic Editor

PLOS ONE

Journal Requirements:

Reviewers' comments:

Reviewer's Responses to Questions

**Comments to the Author**

1. Is the manuscript technically sound, and do the data support the conclusions?

Reviewer #1: Partly

2. Has the statistical analysis been performed appropriately and rigorously? 

Reviewer #1: N/A

3. Have the authors made all data underlying the findings in their manuscript fully available?

Reviewer #1: Yes

4. Is the manuscript presented in an intelligible fashion and written in standard English?

Reviewer #1: Yes

5. Review Comments to the Author

Reviewer #1: This paper investigates the role of heterogeneous susceptibility and infectiousness in a modified SIR compartmental model of disease spread.

I have many concerns about this paper. Most fundamentally, I feel the authors do not demonstrate a deep understanding of the state of the art in mathematical modeling. So it makes me very uncomfortable that there is a thread running through this paper and the other papers of their series in which they criticize modelers, and imply that they have solved everything.

Major concerns:

1) "it became clear that the models were overly pessimistic." This statement in the abstract makes me worry that this is actually highly motivated reasoning masquerading as science. It is good that the authors recognize this claim is not justified - lines 6-7 and 12 of the text contradict the claim in the abstract that this was clear, and acknowledge that no such claim can be made. But now I'm wondering why did they make a claim that they knew was false in the abstract? As a result, at this stage I have significant doubts that I'm about to read a scientific paper. I've read plenty of opinion pieces dressed up in academic language during the pandemic, and I don't want to read one in a legitimate scientific journal.

Related to this, the authors need to get rid of the judgmental statements about existing models and modelers throughout this and the other papers in this "series of articles aimed at better understanding the discrepancy between model prediction and observed data." It is not clear if they actually know the structure of the model whose implied failure is used to motivate the study - someone reading this paper's description would assume that Imperial used a differential-equations based SEIR model that assumed a well-mixed population. That is not what they did.

2) Far too much of this paper is saying "we show in [5] that this cannot be explained without pre-immunity". [5] is a preprint, and I am uncomfortable relying so strongly on results that have not undergone peer review.

3) "The NPI's have remained virtually constant throughout the cold period of 2020-2021... in theory means that the mathematical models should work, just with a new R_0-value adopted to the new situation. However in [5] we make a major effort to produce refinements of state of the art compartmental models, taking various sorts of population heterogeneities into account [and can only fit data by assuming 60% are immune]".

I'm not in a position to comment on the NPIs in Sweden, but when I look at mobility data for Sweden, I see that mobility was highly variable, and that there was a very significant drop in average movement during the second wave, and the drop is likely not uniform in time. Although I have not read [5] nearly as closely as this draft, the claim is made that we can treat behavior as constant because the NPIs are constant. But the measured behavior was not constant, so I think the claim that this can be ignored is false (this is part of the reason I am uncomfortable with such high dependence on a preprint). Additionally, the fact that the behavior changeis likely bimodal (with some individuals significantly reducing movement while others continue as before) is very likely to result in a subpopulation with significantly reduced probability of infection - which will be indistinguishable from a subpopulation having pre-existing immunity.

4) lines 38-40 "In this article we show mathematically that, rather surprisingly, variatio0ns in infectivity has no bearing on the model curves, whereas variations in susceptiblity manifests itself as pre-immunity on the macro level. "

And lines 167-173:

"Incidentally, at the end of each paper [10-12], Kermack and McKendrick stress that a weakness in their model is that they assume uniform susceptibility, which they consider unrealistic in many cases. However, it seems that they never got around to correct this issue, and we have not found a rigorous analysis of how to model this mathematically elsewhere in the literature either. In particular, the formula 1 -1/R0 for the Herd-Immunity Threshold (HIT) (see SM Sec. 4), which stems from their seminal papers, may very well be inaccurate. In the coming section we derive a refined version of this formula taking variable susceptibility into account."

The result of variations in infectivity is not a surprise to me. The variation in infectivity is unimportant under specific circumstances, which do apply in a well-mixed compartmental model settings (and in disease spread in some random networks as well). The reason for this is straightforward: the model assumes that the number of infections is large enough for a law of large numbers to apply (otherwise the model would not be deterministic). As such we can safely average the number of infections caused in a well-mixed population over all individuals. Assuming that infectiousness and susceptibility are uncorrelated, the average infectiousness is the same early or late in the epidemic. (this breaks down if they are biologically correlated or if the higher infectiousness is due to behaviors that also increase susceptibility, in which case the more susceptible are infected sooner on average). [note, if a population is made up of many small groups such as households, then superspreading does have an impact, but this model does not contain households. If the model is intended to predict probability of establishment, then variation is infectiousness (but not susceptibility) matters]

As for variation in susceptibility, the disease preferentially removes the highly susceptible, leaving a less susceptible residual population. There is a large body of work that analyzes this effect. I am surprised that the authors were unable to find a rigorous analysis of this. I would start with:

- Britton, T., Ball, F. & Trapman, P. A mathematical model reveals the influence of population heterogeneity on herd immunity to SARSCoV-2.

- Gou, W. & Jin, Z. How heterogeneous susceptibility and recovery rates affect the spread of epidemics on networks.

-Gerasimov, A., Lebedev, G., Lebedev, M. & Semenycheva, I. COVID-19 dynamics: A heterogeneous model.

-Hickson, R. & Roberts, M. How population heterogeneity in susceptibility and infectivity influences epidemic dynamics.

-Dolbeault, J. & Turinici, G. Social heterogeneity and the COVID19 lockdown in a multi-group SEIR model.

-Miller, J. C. Epidemic size and probability in populations with heterogeneous infectivity and susceptibility

-Miller, J. C. Bounding the size and probability of epidemics on networks

-Miller, J. A note on the derivation of epidemic final sizes

This last one does a lot to explain the issues that the authors raise (heterogeneity in susceptibility mattering but not infectiousness). It also does age-structured models and many other cases. I think it only touches on the final size relations, though likely the arguments used can apply for intermediate times.

There is some subtlety regarding the herd immunity threshold that has not been studied extensively previously. It depends on whether the immunity is achieved through infection or through random vaccination - for the result on infection acquired immunity I would read Gabriela Gomes & colleagues recent papers/preprints. For random vaccination it remains 1-1/R_0.

5) I am concerned by the discussion starting at line 55:

"For example, a SIR-model has no memory, i.e. it does not keep track of how long a person is sick, but the original equation systems in [10] did. However, it has been shown that this, as well as other factors such as randomness has a very limited bearing on the model curves, see e.g. [13]"

Given that the paper starts by referring to the Imperial modeling work and focuses significant attention on implied failure of their predictions, this statement should not be made because it seems to imply that Imperial's (and other models) use a model as in the SIR model described. Most state of the art models (including Imperial's) do have memory.

I don't know that the results from [13] are consistent with this claim. It's not clear what is meant by 'randomness', but definitely some forms of randomness matter and the details of the generation interval plays an important role in the early dynamics.

6) It is not clear to me how system (3) was handled in terms of R_0. Was R_0 defined initially and then a subset treated as immune (so the effective reproduction number drops)? This is what the statement seems to imply. Or was R_0 defined based on the population after the subset was made immune. In either case, this is just the usual SIR model but confined to a smaller population.

Minor:

1) A source is needed for this claim: "most research teams use extensions of SEIR for modeling COVID-19" Additionally the comment needs to distinguish between an SEIR model in terms of progression of infection and the differential equations SEIR model. E.g., the Imperial Model that the authors seem to focus on does not use an SEIR compartmental model, though they assume SEIR progression of infection. Their model is an individual based simulation. It has households, workplaces, ...

2) I don't believe I have encountered the word "fictive" before (google tells me what it is, but I think it is not standard English).

3) It's not appropriate to refer to the epidemic curves as "bell-shaped". That usually implies e^{-x^2} type behavior. These have exponential growth early and exponential decay late.

4) To my eye, most predicted epidemic curves are not symmetric.

5) caption for fig 1: the "second two" do not level out below the HIT. They level out below the HIT predicted by the "standard" model. But that's because they use they are created using a different model. They overshoot the HIT (the HIT is not defined based on a specific canonical model, it's defined based on when R_e=1 for the actual system). Please introduce a notation for the HIT based on the classical model, and discuss it. The discussion of HIT in the context of a different model is confusing because I keep assuming we are discussing the HIT of that model.

6) It is not clear to me why hard-hit towns in Italy which underwent some of the strictest NPIs anywhere would be expected to have high seroprevalence.

7) I am not aware of any serious academic study of "immunological dark matter". I

6. PLOS authors have the option to publish the peer review history of their article (what does this mean?). If published, this will include your full peer review and any attached files.

Reviewer #1: No

---

## [Author Response · Author response to Decision Letter 0]

26 May 2022

Our response to the comments by Reviewer 1 and the editor are found in the attached word document "response to the reviewers". Note that we were unable to get access to the comments of Reviewer 2. We contacted the journal several times about this issue and finally we decided to submit on time. We would of course be happy to see and respond to the comments of Reviewer 2 as well.

---

## [Decision Letter · Decision Letter 1]

23 Nov 2022

PONE-D-22-06269R1A note on variable susceptibility, the herd-immunity threshold and modeling of infectious diseasesPLOS ONE

Dear Dr. Carlsson,

Thank you for submitting your manuscript to PLOS ONE. After careful consideration, we feel that it has merit but does not fully meet PLOS ONE’s publication criteria as it currently stands. Therefore, we invite you to submit a revised version of the manuscript that addresses the points raised during the review process.

We look forward to receiving your revised manuscript.

Kind regards,

Claudine Irles, Ph.D.

Academic Editor

PLOS ONE

Journal Requirements:

Reviewers' comments:

Reviewer's Responses to Questions

**Comments to the Author**

1. If the authors have adequately addressed your comments raised in a previous round of review and you feel that this manuscript is now acceptable for publication, you may indicate that here to bypass the “Comments to the Author” section, enter your conflict of interest statement in the “Confidential to Editor” section, and submit your "Accept" recommendation.

Reviewer #2: All comments have been addressed

2. Is the manuscript technically sound, and do the data support the conclusions?

Reviewer #2: Partly

3. Has the statistical analysis been performed appropriately and rigorously? 

Reviewer #2: Yes

4. Have the authors made all data underlying the findings in their manuscript fully available?

Reviewer #2: Yes

5. Is the manuscript presented in an intelligible fashion and written in standard English?

Reviewer #2: Yes

6. Review Comments to the Author

Reviewer #2: There are some corrections and important to solve few questions. It is also necessary for readers and to improve quality of the manuscript.

7. PLOS authors have the option to publish the peer review history of their article (what does this mean?). If published, this will include your full peer review and any attached files.

Reviewer #2: **Yes: **Md Kamrujjaman

---

## [Author Response · Author response to Decision Letter 1]

4 Dec 2022

We thank the reviewer for insightful comments that helped improve the manuscript. We have revisited the entire text, rewriting bits and pieces in order to make it more readable and understandable. We have taken particular care to figure axes labels, figure titles and figure texts.

All major changes are marked in red in the manuscript titled “with track changes”. Note that text that was slightly rewritten was also marked in red, so there are no essentially new parts.

To address the specific questions by the reviewer, we have clarified that “variable susceptibility” refers to differences between individuals in the probability to get infected, given a meeting with an infective person, and not, as one could get the impression, individual variations in susceptibility over time. We have clarified this in the introduction. We added a headline “novel contributions” to the introduction in order to make this more clear to the reader. The above adresses comments 1-4 by the reviewer. Concerning 5, we have included a number of relevant new citations, among them three of those that the reviewer suggested. We also added newer research indicating that variable susceptibility indeed was present at the onset of the pandemic, see reference [18] Kundu et al, Nature Communications. Also reference [4] by Carlsson and Söderberg-Naucler has in the meanwhile gotten published, so we put more emphasis on this in the discussion on the discrepancy between model output and reality. 

Concerning 5, yes we claim that E plays a minor role, as long as parameter values are free, in the sense that for each parameter configuration for SEIR we can get an almost identical curve using SIR. This is demonstrated in Fig. 3, but the conclusion is well tested on a number of different parameter settings, and this is discussed in depth in Section 4.1. For clarity, these findings have been highlighted also in 2.1, see the new text in red.

Finally, we refrain from adding new graps related to R_0<1, since we feel this would make the manuscript too long and deviate attention from the key findings. Concerning 6, we have carefully read the entire text with a focus on English and punctuation.

---

## [Editor Report · Decision Letter 2]

7 Dec 2022

A note on variable susceptibility, the herd-immunity threshold and modeling of infectious diseases

PONE-D-22-06269R2

Dear Dr. Carlsson,

We’re pleased to inform you that your manuscript has been judged scientifically suitable for publication and will be formally accepted for publication once it meets all outstanding technical requirements.

Kind regards,

Claudine Irles, Ph.D.

Academic Editor

PLOS ONE
---

## [Editor Report · Acceptance letter]

22 Dec 2022

PONE-D-22-06269R2 

A note on variable susceptibility, the herd-immunity threshold and modeling of infectious diseases 

Dear Dr. Carlsson:

I'm pleased to inform you that your manuscript has been deemed suitable for publication in PLOS ONE. Congratulations! Your manuscript is now with our production department. 

Kind regards, 

on behalf of

Dr. Claudine Irles 

Academic Editor

PLOS ONE